# Robotic Liver Resection for Hepatocellular Carcinoma: A Multicenter Case Series

**DOI:** 10.3390/cancers17030415

**Published:** 2025-01-27

**Authors:** Silvio Caringi, Antonella Delvecchio, Maria Conticchio, Francesca Ratti, Paolo Magistri, Andrea Belli, Graziano Ceccarelli, Francesco Izzo, Marcello Giuseppe Spampinato, Nicola De’Angelis, Patrick Pessaux, Tullio Piardi, Fabrizio Di Benedetto, Luca Aldrighetti, Riccardo Memeo

**Affiliations:** 1Department of Surgery, Università Degli Studi Roma “Tor Vergata”, Via Montpellier 1, 00133 Rome, Italy; 2Unit of Hepato-Biliary and Pancreatic Surgery, “F. Miulli” General Hospital, Acquaviva delle Fonti, 70021 Bari, Italy; a.delvecchio@miulli.it (A.D.); m.conticchio@miulli.it (M.C.); r.memeo@miulli.it (R.M.); 3Hepatobiliary Surgery Division, IRCCS San Raffaele Scientific Institute, 20132 Milano, Italy; ratti.francesca@hsr.it (F.R.); aldrighetti.luca@hsr.it (L.A.); 4Hepatobiliary Surgery Division, Vita-Salute San Raffaele University, 20132 Milano, Italy; 5Unit of Hepato-Pancreato-Biliary Surgery and Liver Transplantation, University of Modena and Reggio Emilia, 41121 Modena, Italy; paolo.magistri@unimore.it (P.M.); fabrizio.dibenedetto@unimore.it (F.D.B.); 6Unit of Hepato-Biliary and Pancreatic Surgery, Istituto Nazionale Tumori IRCCS Fondazione G. Pascale, 80131 Napoli, Italy; a.belli@istitutotumori.na.it (A.B.); f.izzo@istitutotumori.na.it (F.I.); 7Unit of General Surgery, San Giovanni Battista Hospital, USL Umbria 2, 06034 Foligno, Italy; graziano.ceccarelli@uslumbria2.it; 8Unit of General Surgery, “Vito Fazzi” Hospital, 73100 Lecce, Italy; marcellogiuseppe.spampinato@asl.lecce.it; 9Unit of Digestive and Hepatobiliary Surgery, Centre Hospitalier Universitaire Henri Mondor, 94000 Créteil, France; nicola.deangelis@unife.it; 10Department of Visceral and Digestive Surgery, University Hospital of Strasbourg, 67000 Strasbourg, France; patrick.pessaux@chru-strasbourg.fr; 11Unit of Surgery, Hôpital Robert Debré, 51100 Reims, France; tpiardi@chu-reims.fr

**Keywords:** robotic surgery, liver resections, robotic hepatectomy, minimally invasive liver surgery, hepatocellular carcinoma, oncological outcome

## Abstract

The success of robotic liver surgery is widespread, and more and more studies are reporting its benefits in terms of surgical and oncological safety. Our study was conducted by analyzing a European multicenter database including 373 robotic liver resections. We analyzed patient and HCC characteristics, data on surgery, the post-operative course, the recurrence rate, the 90-day hospital readmission, and the 90-day mortality rates. We then commented on the results of the analysis by comparing them with those reported in the literature. To our knowledge, this article reports the largest series reported for robotic liver resections.

## 1. Introduction

Liver resection remains, to date, the standard treatment for resectable hepatocellular carcinoma (HCC) [1] and, historically, open liver resection has been considered the gold standard procedure for treating HCC. However, over the years, minimally invasive liver surgery has been associated with a reduction in post-operative pain, post-operative complications, and length of hospital stay, without compromising oncological outcomes. In particular, robotic technology has made it possible to expand the scope and complexity of caries compared to conventional laparoscopy due to improved ergonomics, better three-dimensional visualization, and the application of technology that facilitates the surgical act [2,3]. The literature shows that in appropriately selected patients, RLR is a feasible and safe option when performed by surgeons experienced in open liver surgery [4,5].

To our knowledge, this article reports the largest series reported for RLR.

## 2. Materials and Methods

Data were collected from a multicenter retrospective database that includes 1070 consecutive RLRs performed in nine European hospital centers from 2011 to 2023. Of the entire series, 343 liver resections were performed for resectable HCC. Patients younger than 18 years, with ASA ≥ IV, without portal hypertension, and with non-HCC lesions upon definitive histological examination were excluded from the study. In each center, patients signed an informed consent for the collection of clinical data. We analyzed preoperative, intraoperative, and post-operative data retrospectively.

Preoperative data consisted of patient and HCC characteristics. Specifically, age, sex, BMI, American Society of Anesthesiologists (ASA) score, Charlson comorbidity score, presence of cirrhosis and its etiology, Child–Pugh score, and MELD (Model for End-Stage Liver Disease) score. We also analyzed how many patients had previous surgery and divided them into two groups, open and minimally invasive surgery. On what concerns HCC characteristics, we analyzed the size and location of HCC and alpha-fetoprotein (AFP), and we calculated the TAMPA difficulty index score [6] for all the procedures. Patterns of distribution of multiple lesions were also analyzed. All patients underwent preoperative staging with chest–abdomen–pelvis CT, which also allowed us to define the diameter of the HCC and possible vascular contact. MRI was not routinely used. According to the European Association for the Study of the Liver (EASL) [7], in case of an unclear radiological diagnosis, a liver biopsy was performed. All the clinical cases were discussed by a multidisciplinary team.

All the surgical procedures were performed by expert hepatobiliary surgeons with proven experience in both open and minimally invasive surgery. Liver segmentation anatomy was defined using the Couinaud classification while the Brisbane 2000 terminology was used to define the liver resection type [8].

All procedures were performed with the da Vinci Xi Surgical System. At the beginning of the procedure, an intraoperative ultrasound (IOUS) was systematically performed. Pringle maneuver was routinely prepared at the beginning of the procedure as well, but it was used according to each center’s experience. The parenchymal liver transection was performed with different techniques, energies, and devices depending on the surgeon’s experience and habits.

Data on the type of resection, conversion rate, operative time, estimated blood loss, blood transfusion, and pedicle clamping were recorded. We divided the type of resection into both major and minor resections and anatomical and non-anatomical resections. Regarding the conversion rate, we analyzed the percentage of open and laparoscopic conversions. Estimated blood loss was measured as the difference between the fluid present in the suction receptor and the lavage solution infused into the abdominal cavity.

Post-operative data were recorded, including post-operative complications such as biliary leakage, hemorrhage, prolonged pain ascites, pulmonary infection, and other infections. Severe complications, according to the Clavien–Dindo grading system [9], and reintervention were recorded too.

Data on intensive care unit (ICU) stay, total hospital stay, surgical margins, R1 parenchymal resection, readmission at 90 days, disease recurrence, and mortality at 90 days were recorded.

## 3. Results

Patients’ characteristics and perioperative data are shown in Table 1.

The mean age was 68 years and 78.42% were male patients. The mean BMI was 27 kg/m^2^. A total of 55.97% of patients had an ASA ≥ III. The Charlson comorbidity score mean was 5.73 while the median was 6. One hundred and thirty-five patients (39.65%) had a previous abdominal surgery with similar percentages between open (18.07%) and minimally invasive surgery (21.58%).

As shown in Table 1, 257 patients (74.92%) had liver cirrhosis, with 146 (56.8%) of them being of viral etiology, which is in agreement with literature data showing that in Western nations, the incidence of viral cirrhosis is decreasing in comparison to causes related to metabolic syndromes [10]. The other 86 patients (25.08%) had a value of fibrosis ≥ F3 during the Fibroscan. In total, 320 patients (93.3%%) had a Child–Pugh score of A, and 23 (6.7%) had a Child-Pugh score of B. The mean MELD score was 8.09. The preoperative mean AFP was 270.3 ng/mL and 127 patients (37.03%) had negative values (<7 ng/mL). The mean size of the largest lesions was 33.65 mm. The mean TAMPA score was 1.82 with 68 (19.8%) patients having a score of 1, 273 a score of 2 (79.6%), 1 (0.3%) a score of 3, and 1 (0.3%) a score of 4.

We also analyzed the distribution of HCC in the eight liver segments described by Couinaud. In 72.3% of the cases, the lesions were single, and their distribution was fairly homogeneous across the various liver segments (Figure 1).

In 27.7% of cases, HCC was multifocal, and we analyzed the incidence of involvement of each liver segment in these cases. Data are reported in Figure 2.

Intraoperative data are summarized in Table 2. Major hepatectomies and anatomical resections, defined according to the Brisbane 2000 terminology of liver anatomy and resections [11], were found in 87% and 55% of patients, respectively (Figure 2). In total, 155 (45.2%) wedge resections, 103 (30%) segmentectomies, 40 (11.7%) bisegmentectomies, 31 (9%) left hepatectomies, 11 (3.2%) right hepatectomies, 1 (0.3%) mesohepatectomy, 1 (0.3%) extended left hepatectomy, and 1 (0.3%) associating liver partition and portal vein ligation for staged hepatectomy (ALPPS) were performed. All 17 conversions (4.95%) were to the open approach. The operative mean time was 239.56 min while the median was 220 min. The mean estimated blood loss was 229.45 mL with a median of 150 mL, and fourteen patients (4.08%) had an intraoperative blood transfusion. In 40.23% of cases, pedicle clamping was performed.

The overall post-operative complication rate was 22.74%, and the most represented complications were ascites (10.49%), pulmonary infections (6.41%), prolonged pain (5.83%), and biliary leakages (2.62). Severe complications occurred in 4.08% of patients and one of them (0.29%) was reoperated on. The mean hospital stay was 5.82 days with a mean ICU stay of 0.9 days while the median hospital stay was 5 days. Twenty-six resections (7.6%) were R1 parenchymal. Forty-six patients (4.08%) were readmitted to the hospital within 90 days after discharge and seventy-eight patients (22.74%) had disease recurrence. Total deaths included 36 (10.5%) patients with a 90-day mortality of 0.9%. All post-operative data are summarized in Table 3.

## 4. Discussion

Performing liver resections with a minimally invasive approach has always been challenging due to the depth of the liver’s anatomical locations and its abundant and unique vascularization which have created technical difficulties in safely reproducing the basic maneuvers of open liver surgery laparoscopically [12]. Thanks to the robotic system, it seems possible to overcome some of the limitations of laparoscopic surgery which made this type of approach less safe than open surgery. Consequently, for more complex cases, it was preferred to perform open liver resections, losing all the benefits that minimally invasive surgery guarantees in other types of surgery such as gastrointestinal or urological surgery. In particular, robotic surgery enables stable three-dimensional vision that provides resolution, depth perception, and excellent magnification, a fourth arm that can serve as a stable retractor during all phases of liver resection under the surgeon’s direct control, and wrist-worn instruments that enhance the suturing and knotting capabilities of laparoscopy, which is very useful for vascular control and hemostasis. Together, the use of increasingly advanced technologies applicable to robotic surgery makes this type of approach increasingly safe from a surgical and oncological point of view.

In this case series, all resections were made for resectable HCC in adults with compensated cirrhosis without portal hypertension and with an acceptable operative risk according to the ASA score. On average, patients were elderly, according to the World Health Organization definition, male, and overweight [13] with an average 85% one-year mortality according to our data regarding mean Charlson comorbidity index score [14]. A large part of the sample had already performed previous abdominal surgery, and it is known that this can complicate the surgical procedure due to the presence of abdominal adhesions. However, as already described in some works reported in the literature [15,16], this did not impact post-operative complications or operating times as it does in laparoscopic surgery, demonstrating how robotic surgery can combine the manual ability of open surgery with minimal invasiveness. We also analyzed the alpha-fetoprotein values and noticed how the mean and median values are certainly above the maximum cut-off described in the literature of 5–10 ng/mL [17] but, at the same time, are lower than the minimum cut-off of 400 ng/mL that the literature considers diagnostic for HCC [18]. Our data, therefore, are consistent with some studies that demonstrate how a good proportion of HCCs actually have normal AFP values [19], suggesting the need to identify more sensitive and specific biomarkers for the early diagnosis of HCC.

On average, resected HCCs presented in stage II or III according to TNM staging [20], and most of them were mono-focal lesions with a uniform distribution in the liver segments. Our data are therefore in contrast with what is reported in the literature, according to which the distribution of HCCs in the various liver segments reflects their volume, responding to probabilistic laws [21,22].

The larger volume liver segments such as sVIII are also those with greater difficulty in execution as indicated by the Tampa difficulty score. According to our data, the average Tampa difficulty score was 1.82 with a median of 2, indicating that resections belonging to group 1 (less demanding) and group 2 (intermediate) were more frequent. In this case series, therefore, our reported incidence of HCC by liver segment does not disagree with what is reported in the literature [21,22]. There is a bias due to the difficulty of performing the surgery in those hepatic segments with larger volumes. Although robotic surgery overcomes some of the limitations of laparoscopic surgery by making the execution of some procedures easier, some surgical interventions with a high difficulty of execution are still preferentially performed with an open approach.

To support this, only 13% of the operations performed were major hepatectomies and 45% were non-anatomic resections. Taking a closer look at the group of major hepatectomies, those of greater difficulty are fewer, demonstrating how, in these cases, an open approach is probably preferred.

However, in the sample under examination, there were tricky cases, and in seventeen cases (4.95%), conversion was necessary, nine times for anesthesiologic problems and eight times due to oncological radicality. All conversions were to open surgery. The data agree with the literature and, in particular, as reported by Montalti et al., the percentage of conversions to open surgery is lower in robotic surgery than in laparoscopic surgery [23,24]. The reason probably lies in what was said previously: robotic liver surgery, performed by expert surgeons, allows to both better perform complex resections and control intraoperative adverse events. It is no coincidence, in fact, that all the conversions were to open surgery.

The data concerning the duration of surgery and blood loss are similar to what has been reported in other significant studies [25,26]. It is common knowledge that the perioperative preparation and logistics of robotic surgery, including anesthetic strategies, increase the operative time. Only one paper described and analyzed the data regarding the docking time and console time [27]. Therefore, while it is true that surgical times are shorter in open surgery than in robotic surgery, on the other hand, the advantage of robotic surgery is more evident in the post-operative period. Almost all post-operative complications require only pharmacological treatment and not surgical or endoscopic reinterventions or interventional radiology, which is relevant when treating patients with cirrhosis who are at greater risk of hepatic decompensation after surgery. Specifically, only in one case has it been necessary to return to the operating theater, and the most frequent post-operative complication was ascites, a condition that is, however, as frequent as it is not clinically impactful in cirrhotic patients of whom the study sample consists. We can therefore confirm what is now well established in the literature, i.e., that robotic liver surgery, performed by experienced surgeons in high-volume centers, has a lower rate of post-operative complications, allowing a shorter hospital and ICU stay than open liver surgery, at the price of a longer duration of the operation. It is well known that one of the major criticisms of the use of robotic technology in surgery is the cost, but the reduction in ICU and hospital time and the lower incidence of serious complications lead to a reduction in overall costs and could compensate for the higher expenses associated with the technology. We could not perform a cost-effectiveness analysis due to the different reimbursement systems and uneven costs in the various countries of the participating centers. However, a median difference of 6 days of hospitalization and a reduction in the need for ICUs is reported in the literature [26] and this, intuitively, indicates that robotic liver surgery represents major cost savings for any hospital. Regarding safety, it must also be considered that, at ninety days after surgery, only ten of the fourteen patients readmitted to the hospital had post-operative problems while the other four were readmitted for other non-clinical reasons. Ninety-day mortality was also low, with only three cases (0.9%), all due to liver failure.

Robotic liver surgery has also proven to be effective and safe from an oncological point of view. The data from our sample regarding the distance of HCC from the margins, as well as the frequency of R1 and disease recurrence, not only agree with the other data reported in the literature but, in some cases, are even better [28,29,30,31].

## 5. Limitations

Limitations of this study include the retrospective design and potential biases of a multicenter, nonrandomized protocol.

## 6. Conclusions

To the best of our knowledge, this case series represents the Western’s largest case series on RLRs, and its results confirm what other studies reported in the literature suggest. RLR for HCC confers significant advantages with excellent safety profiles when performed in experienced centers by surgeons who have completed the learning curve. In particular, robotic liver surgery may reduce morbidity by expanding the potential number of patients able to receive treatment from which they are currently excluded due to the risk of liver decompensation [26]. The only limitation of robotic liver surgery, but indeed of all robotic surgery, is the cost. It is, therefore, necessary to produce a study on cost-effectiveness that can analyze the costs of robotic liver surgery and confirm the intuition that its positive effects can mitigate the exorbitant costs of the technology used.

## Figures and Tables

**Figure 1 cancers-17-00415-f001:**
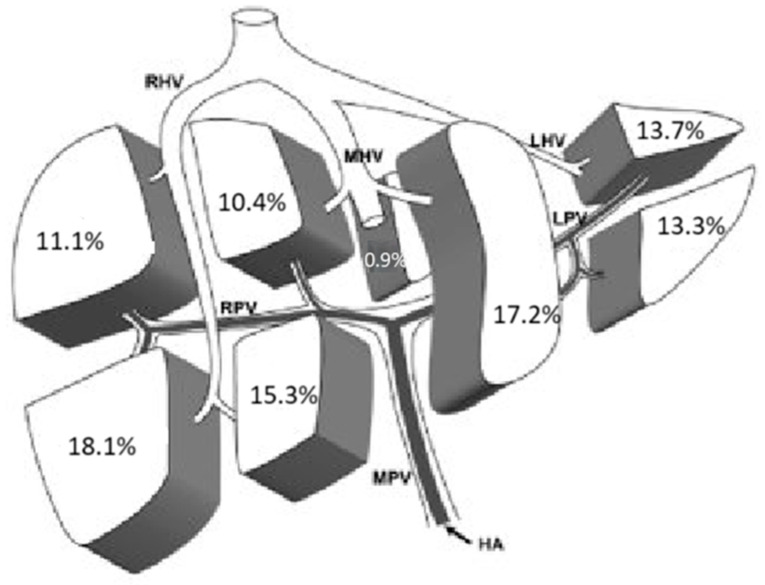
Localization of single lesions. HA: Hepatic Artery; MPV: Main Portal Vein; RPV: Right Portal Vein; LPV: Left Portal Vein; RHV: Right Hepatic Vein; MHV: Middle Hepatic Vein; LHV: Left Hepatic Vein.

**Figure 2 cancers-17-00415-f002:**
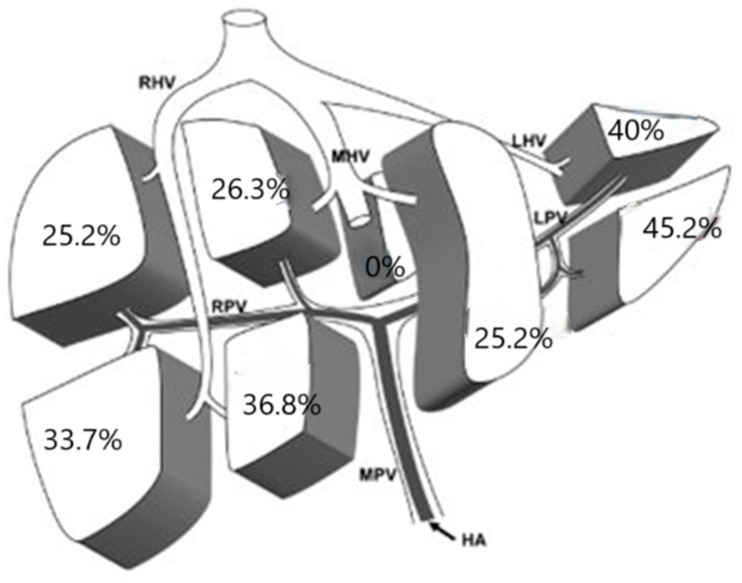
Localization of multiple lesions. HA: Hepatic Artery; MPV: Main Portal Vein; RPV: Right Portal Vein; LPV: Left Portal Vein; RHV: Right Hepatic Vein; MHV: Middle Hepatic Vein; LHV: Left Hepatic Vein.

**Table 1 cancers-17-00415-t001:** Patients’ characteristics and perioperative data.

Variables	*n* = 343
Age (yr), mean	68
Male, *n* (%)	78.42%
BMI (kg/m^2^), mean	27
ASA score ≥ III, *n* (%)	55.97%
Charlson comorbidity score, mean	5.73
Previous open abdominal surgery, *n* (%)	18.07%
Previous laparoscopic abdominal surgery, *n* (%)	17.49%
Cirrhosis, *n* (%)	74.92%
Viral etiology, *n* (%)	56.8%
MELD Score, mean	8.09
Child–Pugh score, *n* (%)	A (93.3%), B (6.7%)
Number of lesions, mean	1.26
Tumor: size of the biggest lesion (mm), mean	33.65
TAMPA score, mean	1.82
Alpha-fetoprotein, mean–median	270.30–270

BMI: body mass index; ASA: American Society of Anesthesiologists; MELD: mayo end-stage liver disease.

**Table 2 cancers-17-00415-t002:** Intraoperative data.

Variables	*n* = 343
Major resection, *n* (%)	13.12%
Anatomical resection, *n* (%)	54.8%
Conversion to open, *n* (%)	4.95%
Conversion to laparoscopy, *n* (%)	0
Operative time (min), mean–median	239.56–220
Estimated blood loss (mL), mean–median	229.45–150
Blood transfusion, *n* (%)	4.08%
Pedicle clamping, *n* (%)	40.23%

**Table 3 cancers-17-00415-t003:** Post-operative data.

Variables	*n* = 343
Post-operative complication, *n* (%)	22.74
Biliary leakage, *n* (%)	2.62
Hemorrhage, *n* (%)	2.33
Prolonged pain, *n* (%)	5.83
Ascites, *n* (%)	10.49
Pulmonary infection, *n* (%)	6.41
Other infections, *n* (%)	5.25
Severe complication (Clavien ≥ 3), *n* (%)	4.08
Reintervention, *n* (%)	0.29
ICU stay (days), mean	0.90
Total hospital stay (days), mean–median	5.82–5
Surgical margins (mm), mean-median	10.07–10
R1 parenchymal, *n* (%)	7.6
Readmission at 90 days, *n* (%)	4.08
Disease recurrence, *n* (%)	22.74
Mortality at 90 days, *n* (%)	0.9

ICU: intensive care unit.

## Data Availability

The data presented in this study are available upon request from the corresponding author due to privacy reasons.

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
