# Peer review of "Robotic Liver Resection for Hepatocellular Carcinoma: A Multicenter Case Series"

_cancers, 2025, doi:10.3390/cancers17030415_

Round 1
Reviewer 1 Report
Comments and Suggestions for Authors
The authors focused on reporting about robotic liver resection for hepatocellular carcinoma: a multicenter case series. The paper is very readable. However, the several following issues should be reflected.
1) I think these research results lack some degree of novelty.
2) What is the model of the robot?
3) What were the causes of the cases that required conversion to open surgery?
4) It is stated in the conclusion as follows. In particular, robotic liver surgery may reduce morbidity by expanding the potential number patients able to receive treatment from which they are currently excluded due to the risk of liver decompensation. Which results from this study support this suggestion?
Author Response
Comments 1: I think these research results lack some degree of novelty.
Response 1: Thank you for pointing this issue out. The results of our research confirm the insights described in the literature by some comparative studies. However, it is proposed as the largest case series in the Western world, giving a push for new and more robust studies.
Comment 2: What is the model of the robot?
Response 2: Thank you for pointing this issue out. I included within the methods the robot model used, which was the da Vinci Xi in all centres
Comment 3: What were the causes of the cases that required conversion to open surgery?
Response 3: I agree with you. The causes of conversion to open surgery are described within our database but were deliberately not specified and analysed in our study so as not to make it too elaborate and mislead the object of our research. However, we agree that it is a very interesting topic for discussion and will therefore be the subject of a further article, currently being written.
Comment 4: It is stated in the conclusion as follows. In particular, robotic liver surgery may reduce morbidity by expanding the potential number patients able to receive treatment from which they are currently excluded due to the risk of liver decompensation. Which results from this study support this suggestion?
Response 4: Thank you for pointing this issue out. We forgot to include the reference of this suggestion.
Reviewer 2 Report
Comments and Suggestions for Authors
I applaud the authors Caringi et all on their wonderful multicenter case series of robotic liver resection. This is valuable and I believe this should be published.
I have only two comments which are minor.
1. The major addition of robotic liver resection should be placed in context of the already existing MIS approaches. Indeed there have already been studies comparing robotic and laparoscopic liver resection, including one randomized study (PMID 39210947) and retrospective data (PMID 35641702). They might additionally compare robotic to open resection (PMID: 37620660) in the effort of showing how far we are progressing in the field. Finally, they might compare to benchmark outcomes of laparoscopic surgery from Goh 2023 PMID 35837974
2. The question of resection vs transplant remains very relevant. The authors very nicely describe their inclusion criteria but I would love to hear their thoughts on why transplant is not pursued. Theoretically this leaves cirrhotic liver behind, even if compensated. This is particularly relevant with increasing graft availability with machine perfusion (PMID 38833290, 38557793). There have even been calls for expanding selection criteria now for HCC given the equivalent outcomes after transplant and improved graft availability. Curious thoughts on this.
Overall this is a great series and should be published.
Author Response
Thank you for your review, very appreciated from us all. Thank you also for pointing this issues out. Your two comments are very interesting and will certainly be an inspiration for future studies.